# Repurposing Disulfiram (Tetraethylthiuram Disulfide) as a Potential Drug Candidate against *Borrelia burgdorferi* In Vitro and In Vivo

**DOI:** 10.3390/antibiotics9090633

**Published:** 2020-09-22

**Authors:** Hari-Hara S. K. Potula, Jahanbanoo Shahryari, Mohammed Inayathullah, Andrey Victorovich Malkovskiy, Kwang-Min Kim, Jayakumar Rajadas

**Affiliations:** 1Biomaterials and Advanced Drug Delivery, Stanford Cardiovascular Pharmacology Division, Cardiovascular Institute, Stanford University School of Medicine, Palo Alto, CA 94304, USA; hspotula@stanford.edu (H.-H.S.K.P.); jahansh@stanford.edu (J.S.); inayath@stanford.edu (M.I.); kwangmin@stanford.edu (K.-M.K.); 2Department of Plant Biology, Carnegie Institute for Science, Stanford University, Palo Alto, CA 94305, USA; amalkovskiy@carnegiescience.edu

**Keywords:** lyme disease, *Borrelia burgdorferi*, antimicrobial activity, disulfiram, lyme carditis

## Abstract

Lyme disease caused by the *Borrelia burgdorferi* (*Bb or B. burgdorferi*) is the most common vector-borne, multi-systemic disease in the USA. Although most Lyme disease patients can be cured with a course of the first line of antibiotic treatment, some patients are intolerant to currently available antibiotics, necessitating the development of more effective therapeutics. We previously found several drugs, including disulfiram, that exhibited effective activity against *B. burgdorferi*. In the current study, we evaluated the potential of repurposing the FDA-approved drug, disulfiram for its borreliacidal activity. Our results indicate disulfiram has excellent borreliacidal activity against both the log and stationary phase *B. burgdorferi sensu stricto B31 MI*. Treatment of mice with disulfiram eliminated the *B. burgdorferi sensu stricto B31 MI* completely from the hearts and urinary bladder by day 28 post infection. Moreover, disulfiram-treated mice showed reduced expressions of inflammatory markers, and thus they were protected from histopathology and cardiac organ damage. Furthermore, disulfiram-treated mice showed significantly lower amounts of total antibody titers (IgM and IgG) at day 21 and total IgG2b at day 28 post infection. FACS analysis of lymph nodes revealed a decrease in the percentage of CD19+ B cells and an increase in total percentage of CD3+ T cells, CD3+ CD4+ T helpers, and naive and effector memory cells in disulfiram-treated mice. Together, our findings suggest that disulfiram has the potential to be repurposed as an effective antibiotic for treating Lyme disease.

## 1. Introduction

Lyme disease, a Zoonosis, is the most commonly reported vector-borne disease in the United States and approximately affects 300,000 individuals annually in North America [1] and is spread by the spirochete *Borrelia burgdorferi* sensu stricto (hereafter termed *B. burgdorferi or Bb*). The clinical manifestations of Lyme disease include three phases [2]. Early infection involves localized erythema migrans, followed within days or weeks by dissemination to the nervous system, heart, or joints. Without antibiotics treatment, 60% of patients with Lyme disease in the United States develop arthritis, which may recur at intervals and last for months or years. A fewer number of patients (4 to 10%) suffer carditis, which is an early and nonrecurring manifestation of the infection [3]. The antibiotic treatment using oral doxycycline is effective for most patients at the early localized stage of Lyme disease [4]. Several studies indicate that disseminated infection is not eradicated by conventional antibiotics such as tetracycline, doxycycline, amoxicillin, or ceftriaxone in animal models such as mice [5,6,7], dogs [8], ponies [9], and non-human primates [10,11]. Several reports also showed that antibiotics daptomycin and cefoperazone in combination with doxycycline or amoxicillin effectively eliminated *B. burgdorferi* persisters [12,13]. However, these antibiotic combinations failed to act against *B. burgdorferi* biofilm forms [13]. Macrolides are also less effective against *B*. *burgdorferi*, and resistance of these spirochetes to erythromycin has been reported [14]. As a result, macrolides including azithromycin, clarithromycin and erythromycin are recommended only for patients who are intolerant to the first-line therapy. Therefore, based on these observations, new mechanistic classes of antibiotics need to be developed to treat infections arising from various forms of *B. burgdorferi*. Hence, the discovery of new antimicrobials that could be used alone or in combination with other antibiotics will be highly beneficial for drug-intolerant patients and potentially for patients suffering from chronic Lyme disease that is refractory to other agents.

One approach to expedite the development of new antibiotics is to repurpose preexisting drugs that have been approved for the treatment of other medical conditions. Previously, we screened drugs (80% of them are FDA approved, with a total of 4366 chemical compounds from four different libraries) with high efficacy against the log and stationary phase of *B. burgdorferi* by BacTiter-Glo™ Assay. Amongst them, disulfiram (Antabuse^TM^), an oral prescription drug for the treatment of alcohol abuse since 1949, was found to have the highest anti-persister activity against *B. burgdorferi* [15]. In addition, disulfiram and its metabolites are potent inhibitors of mitochondrial and cytosolic aldehyde dehydrogenases (ALDH) [16]. Recent U.S. clinical trials using repurposed disulfiram treatments include: methamphetamine dependence (NCT00731133); cocaine addiction (NCT00395850); melanoma (NCT00256230); muscle atrophy in pancreatic cancer (NCT02671890); HIV infection (NCT01286259); a modulator of amyloid precursor protein processing (NCT03212599); and a recently initiated for Lyme disease treatment (NCT03891667) [17]. In the area of infectious diseases, disulfiram has been shown to have antibacterial [18,19] and anti-parasitic [20] properties. A recent study showed that disulfiram has beneficial effects in the treatment of patients with Lyme disease and babesiosis [21]. Disulfiram is an electrophile that readily forms disulfides with thiol-bearing substances. *B. burgdorferi* possesses a diverse range of intracellular cofactors (e.g., coenzyme A reductase) [22], metabolites (e.g., glutathione) and enzymes (e.g., thioredoxin) [23] containing thiophilic residues that can be modified by disulfiram through thiol-disulfide exchanges to evoke antimicrobial effects. Therefore, disulfiram has the potential to inhibit *B. burgdorferi* metabolism by forming mixed disulfides with metal ions [24] and our group previously showed that *B. burgdorferi* requires zinc and manganese as co-factors for key biological processes [25]. Recently, we have shown that the drug azlocillin can be effective in controlling doxycycline-persisters of *B. burgdorferi sensu stricto JLB31* in both in vitro and in vivo studies [26].

In the present study, we evaluated the antibacterial activities of disulfiram against log- and stationary phase cultures of *B. burgdorferi sensu stricto B31 MI*. Furthermore, the bactericidal activity of disulfiram in vivo was determined using the C3H/HeN mouse model of Lyme disease at days 14 and 21 *B. burgdorferi sensu stricto B31 MI* infection, a timeline that denotes the early onset of chronic infection.

## 2. Results

### 2.1. Disulfiram Is a Potential Antibiotic on Log Phage and Stationary Cultures of B. burgdorferi B31 MI 

Whereas our initial screen of disulfiram from four drug libraries is based on Bac titer-Glo assay [15], we followed up on these studies and performed our preliminary assays using varying concentrations of disulfiram ranging from 100 µM to 0.625 µM by Bac titer-Glo assay, which measures the cell viability based on quantitation of ATP present, but it cannot discriminate between the inhibitory or bactericidal effects of the disulfiram. Furthermore, we have performed the MICs/MBCs by microdilution assay, a gold standard for measuring the bactericidal effects and also complemented these studies using the morphological evaluation methods like darkfield direct cell counting and SYBR® Green I/PI (Live/Dead) fluorescence microscopy-based counting formats (Figure 1 and Figure 2 and Appendix A). We also considered the effectiveness of the drugs prepared in various solvents including DMSO (Dis-DM), cyclodextrin (Dis-CD) and included doxycycline (Dox) for comparative analysis of the sensitivity of spirochete and the round body morphological forms of *B. burgdorferi* B31 MI in-vitro to disulfiram treatment and included appropriate controls.

Our in vitro studies indicated that the treatment concentrations ranging from 20 µM, 10 µM, 5 µM, 2.5 µM, 1.25 µM and 0.625 µM of disulfiram in DMSO and disulfiram in cyclodextrin drugs significantly eliminated 80–94% of log-phase spirochetes and 78–92% of stationary phase persisters compared to the controls (Figure 1 and Figure 2 and Appendix A). More specifically, treatment with disulfiram at 5 µM (1.48 µg/mL) concentration in DMSO, and cyclodextrin significantly eliminated 94% of log-phase spirochetes. Furthermore, treatment with 5–10 µM (1.48–2.97 µg/mL) concentration of disulfiram in DMSO and disulfiram in cyclodextrin drugs also significantly eliminated 92% and 90% respectively stationary phase spirochetes (Figure 2A,B). Consistent with our earlier study [12], doxycycline significantly reduced the viability of log phase *B. burgdorferi* by 90–97% compared to the control treatment (Figure 1A,B). Furthermore, doxycycline treatment also had a significant effect (around 80 to 85%) on the cells in the stationary phase cultures as observed by decreasing proportion of viable cells after antibiotic exposure compared to the control group (Figure 2B,C and Appendix A). However, at high concentration (ranging from 50–100 µM) they lose efficacy and show reduced bactericidal activity with an increase in the concentration of disulfiram drug in DMSO or cyclodextrin. These results are specific since treatment with ultra-pure water or cyclodextrin or DMSO in BSK medium did not significantly reduce the viability of the spirochete rich log phase culture and persisters rich stationary cultures compared to the controls, observed in both direct dark field SYBR Green-I /PI based fluorescent microscopy counting and semisolid plating methods (Figure 1A and Figure 2A and Appendix A).

To confirm our dark microscopy results, *B. burgdorferi* B31 MI log and stationary forms were further evaluated in vitro for their sensitivity to disulfiram-in-DMSO and disulfiram-in-cyclodextrin by fluorescent microscopy counting method using SYBR Green-I (live cells stain green) and Propidium Iodide (dead cells stain red) and by a semisolid plating method. Consistently, the 5–10 µM (1.48–2.97 µg/mL) concentration of disulfiram-in-DMSO and disulfiram-in-cyclodextrin drugs significantly reduced log phase spirochetes by 94%, but in the remaining 6% of the population, 4% were stained green for alive while 2–3% were stained red for dead (Figure 2B). While doxycycline significantly reduced the viability of log-phase cultures by 97%, it also reduced the stationary phase viability of *B. burgdorferi* by around 80 to 85% (Figure 2B). However, most interestingly treatment with 5–10 µM (1.48–2.97 µg/mL) concentration of disulfiram-in-DMSO and disulfiram-in-cyclodextrin drugs significantly eliminated stationary phase spirochetes by 92% and 90% respectively, but in the remaining 8% and 10%, 6% and 8% were live while 2–4% were dead (Figure 2B). These results were found to agree with the findings from the dark field microscopy counting assays (Figure 2B).

At low concentrations (ranging from 10–0.625 µM), the disulfiram in DMSO and disulfiram in cyclodextrin drugs concentration-response profile is sigmoidal. In contrast, at higher concentrations (ranging from 25–100 µM), the disulfiram drugs lost their efficacy and exhibited a U- or bell-shaped curve as shown in Figure 1 and Figure 2. We attribute this loss in drug efficacy to inadvertent effects arising from the colloidal forms of these drugs at high concentrations. To understand this further we have used the Dynamic light scattering (DLS) and Atomic force microscopy (AFM) based techniques.

### 2.2. DLS and AFM Imaging Analyses Demonstrate the Formation of Disulfiram Aggregates at High Concentrations

The dynamic light scattering (DLS) technique was used to study the effect of concentration of the aggregation of disulfiram. Our data from the Appendix A indicates a variation in the average count rate of the particles with increasing concentrations of disulfiram prepared from DMSO and CD stock solutions. The samples from DMSO preparation showed a linear increase in the average count rate proportional to the concentration. However, the slope of the increase has changed at 10 µM or higher concentrations indicating a critical aggregation concentration (CAC) (Appendix A). The results from the CD preparation showed a non-linear trend with a break at 10 µM (where the red dotted line joins the green dotted line) consistent with the CAC results from DMSO preparation. To study the stability of the aggregates, we have performed a disruption of aggregation study in the presence of standard detergent such as SDS, and DMSO at a high concentration of DSF (well-above CAC). We found that DSF aggregates are stable at 20 mM SDS and up to 40% DMSO and could not disrupt or disaggregate DSF (Appendix A).

Atomic force microscopy-based techniques helped us to further evaluate the small volume (10 µL) of liquid sample aliquots. In this method, fast drying of the highly spread droplets on hydrophilic substrates allowed us to assess the real disulfiram particle dimensions with minimal contribution of secondary sample aggregation due to local increase in its concentration due to drying. As shown in Appendix A, we observed very few aggregates from DMSO samples at all concentrations tested (Appendix A). The smaller particles are less than 1 nm in diameter. For cyclodextrin particles, we could see larger particles that are quite wide, but also quite flat—only 10 nm high, on average (Appendix A). We reasoned that these particles may have formed due to sample drying. However, for smaller particles, a crossover can be seen from 25 to 10 µM (Appendix A). In the former sample, we can still observe them (white arrows), but not in the latter, which is only 2.5 times less concentrated. We believe this provides sufficient evidence for sample aggregation, which appeared to be substantial only above 20 µM, as evident from our DLS data. These results indicate that at higher concentration disulfiram-in-DMSO and disulfiram-in-cyclodextrin drugs form aggregates and losses its borreliacidal activity.

### 2.3. Disulfiram Treatment Reduces the B. burgdorferi Burden in Tissues Following Dissemination in Infected C3H/HeN Mice

Following-up on the potent in vitro killing activity of disulfiram against *B. burgdorferi,* we examined its efficacy in vivo immunocompetent C3H/HeN [27] mice and compared with doxycycline. To better compare the efficacy, disulfiram (75 mg/kg of bodyweight) was introduced intra-peritoneally on day 14 and day 21 (to consistently develop persistent infection and carditis [28]) in to post infected C3H/HeN mice at 75 mg/kg of body weight every day for 5 days (Figure 3A). Mice were sacrificed after 48 h of the last dose to collect tissue samples at both time points (day 21 and day 28 post drug treatment). A whole ear was used for cultivation and analyzing the pathogen loads. The bladder, ear and heart were collected for whole-DNA extraction and Q-PCR analysis (Table 1 and Figure 3A). The *flaB* gene PCR positivity represents either live *Bb* or components that had not been cleared from tissues. In the day 14 post disulfiram treatment group, 3 out of 5 mice were *flaB* gene PCR positive from the heart, bladder and ear tissues (Table 1 and Figure 3B). Similarly, in the doxycycline treatment (50 mg/kg) group, 3 out of 5 mice were positive by PCR from the heart and ear tissues but none were positive for PCR from bladder tissues (Figure 3B). The disulfiram treatment group had a statistically significant lower number of *B. burgdorferi* compared to untreated infected controls (Table 2 and Figure 3B). On the other hand, in the day 21 post disulfiram treatment group, 2 out of 4 mice were *flaB* gene PCR positive from the ear and the rest of the tissues, heart and bladder were PCR negative (Figure 3B). Whereas in the doxycycline treatment group, 4 out of 4 mice ears were PCR positive and 1 out of 4 mice bladder was PCR positive but none were positive for PCR from the heart (Table 1 and Figure 3B). However, no borrelia was observed in the cultured ear of post disulfiram or doxycycline treatment groups. These results showed disulfiram similar to doxycycline in C3H/HeN mouse to restrict the further growth and dissemination of *B. burgdorferi*.


### 2.4. Disulfiram Treatment Decreases Disease Pathology and Further Reduces Inflammatory Markers in the Heart of B. burgdorferi Infected C3H/HeN Mice

In Lyme borreliosis, heavy inflammatory infiltrates dominated by mono or polymorphonuclear leukocytes are typically found at lesion sites [29]. We performed histopathology analysis of the heart in both the day 14 and day 21 post disulfiram treatment group mice and found normal features of the aorta, valves and few to no mononuclear leukocytes inflammation in the myocardium which signifies inactive carditis than the infected mice (Figure 3C and Appendix A), whereas the doxycycline treatment group mice, specifically in day 21, showed mild to moderate level of mononuclear leukocytes inflammation in the aorta and valves which signifies active carditis (Figure 3C and Appendix A) than the infected untreated mice which showed transmural infiltration of mononuclear leukocytes in the aorta and valves signifies severe active carditis (Figure 3C and Appendix A). Overall, these results show that the disulfiram is as effective as doxycycline in decreasing the disease pathology of the infected hearts.

In infectious diseases, a hallmark of inflammatory tissue reactions is the recruitment and activation of leukocytes. Chemo and cytokines play a pivotal role in mediating these events. We further determined whether disulfiram treatment alters the inflammatory responses in the heart at both day 14 and day 21 post infection, we evaluated *B. burgdorferi* induced myocardial inflammation by quantification of mRNA transcripts of CxCL1 (KC), CxCL2 (MIP-2), CCL5 (RANTES), TNF, IFNγ, IL-10, IL-1β, iNOS/NOS-2 by qRT-PCR. In the day 14 post disulfiram treatment group, levels of MIP-2, TNF, IFNγ and IL-10 were significantly lower (reached normal levels) relative to infected untreated mice (Figure 4). More specifically, IL-10 levels were reduced to 60-fold (Figure 4C), while MIP-2, TNF, and IFNγ were reduced to 10-fold (Figure 4A,D,E). There was no change in NOS2 or iNOS levels (Figure 4G,H). On the other hand, in the day 21 post disulfiram treatment group, levels of MIP-2, RANTES, TNF, IFNγ, IL-1β, and IL-10 were significantly lower (touched to normal levels) relative to infected untreated mice (Figure 4), while iNOS and NOS-2 levels were significantly higher relative to infected untreated mice (Figure 4G,H). More specifically, IL-10 levels were reduced to 100-fold (Figure 4C), and other cytokines like MIP-2, TNF, IL-1β, and IFNγ levels were reduced ten to sixty-fold (Figure 4A,D–F). While NOS-2 or iNOS, which have a role in immune regulation and tissues remodeling, were significantly higher compared to infected untreated mice (Figure 4G,H). These results indicate that disulfiram affects the regulation and/or balance of Th1 (MIP-2, RANTES, TNF, IFNγ and IL-1β), Th2 (IL-10) and protective Macrophage M1 (NOS2 or iNOS) responses to *B. burgdorferi* at day 21 and day 28 post infection. However, the doxycycline treatment group have only reduced few cytokines like IL-10, TNF, and IFNγ at day 14 and 21 post treatment (Figure 4C–E). While NOS-2 levels elevated at day 21 and MIP-2 levels reduced at day 28 post infection (Figure 4G,H).

### 2.5. Disulfiram Treatment Reduces Antibody Titers in the B. burgdorferi Infected Mice

We next sought to determine whether disulfiram treatment affects antibody development during day 14 and day 21 post *B. burgdorferi* infection, we measured the serum levels of Bb-specific and total immunoglobulins using an ELISA method. The results on day 21, showed that the Bb specific IgM and the total amount of IgM and IgG levels were significantly lower in disulfiram treated mice compared infected control mice (Figure 5A). Among the total IgG subtypes, total IgG1 levels were significantly lower than the infected control mice (Appendix A). However, there was no effect on other IgG subtypes like IgG2a, IgG2b and IgG3 (Appendix A). On the other hand, at day 28, Bb specific IgM and IgG levels were significantly lower in disulfiram treated mice when compared to those in infected control mice (Figure 5B) while at day 28, only a trend towards lower total IgG levels was observed (Figure 5B). However, IgG2b levels were significantly lower in disulfiram treated mice and there was no effect on other total IgG subtypes like IgG1, IgG2a and IgG3 (Appendix A). Doxycycline treatment group does not show any reduction of total antibody titers at day 21 and day 28 (Figure 5A,B). However, at day 28, doxycycline treatment group showed significantly lower levels of Bb-specific IgM and IgG. These data suggest that the disulfiram treatment might induced development of antibody subtypes very efficiently and affect IgG class switching, which may represent a contributing factor in lowering *B. burgdorferi* titers at day 21 and clearance of *Bb* more efficiently at day 28. However, we cannot exclude the fact that there is a possibility that B cells expressing different immunoglobulin isotypes are selectively expanded.

### 2.6. Disulfiram Reduces Lymphoadenopathy in B. burgdorferi Infected C3H/HeN Mice

Lymphadenopathy, a hall mark of acute Lyme borreliosis [30] manifestation is characterized by increased cellularity and the accumulation of large pleomorphic IgM- and IgG- positive plasma cells. To determine whether disulfiram treatment reduces the lymph node enlargement, on day 28 we collected peripheral (axillary, brachial, cervical and inguinal) lymphnodes (pLNs) and determine the cell number counts followed by analyzing B and T cell populations by flow cytometry. In disulfiram treatment mice, total lymphocytes of pLNs were statistically reduced in comparison to infected control mice (Figure 6). Doxycycline-treated mice also showed similar results. Further, our pLNs FACS analysis of the disulfiram treatment group had a significant decrease in the percentages of CD19+ B cells, and a significant increase in the percentages of CD3+ T cells in comparison to infected control mice (Figure 6). Furthermore, among the CD3+ subsets, CD3+ CD4+ helper T cells and CD3+ CD8+ cytotoxic T cells were not affected in comparison to infected control mice (Figure 6). However, when we compare naïve uninfected mice with all three infected groups (infected PBS treated, infected doxycycline treated and disulfiram treated) showed a significant decrease in the percentages of CD3+ CD8+ cytotoxic T cells (Figure 6), and a significant increase of the percentages of CD3+ CD4+ helper T cells (Figure 6). Another hallmark of effective and long-term protection is the generation of memory T cells. They provide an efficient immune response to pathogen re-exposure [31]. We further analyzed CD4^+^ T helper subsets by labeling naïve (CD62L^+^), early effector (CD62L^−^/CD44^−^), effector (CD44^+^) and memory T cells (CD62L^+^/CD44^+^). Analysis of helper T cells in comparison to naïve uninfected mice revealed that disulfiram treatment mice led to a significant increase in early effector/effector and memory T cells and to a significant decrease of naïve T cells in pLNs (Figure 6). A similar trend was observed in infected PBS treated and infected doxycycline treated mice (Figure 6).

## 3. Discussion

Since antibiotics are the top-of-the-line options to treat infections, there remains a dire need and a practical approach to bring more efficient antibiotics to the clinic. The repurposing of FDA approved antibiotics through fast-track approvals can be an excellent solution. In the current study, we evaluated the borreliacidal potential of an FDA approved drug, disulfiram, using in vitro and in vivo experimental models based on our previous high-throughput screening hits [15,32]. We performed preliminary in vitro antimicrobial assays by Bac-titer Glo assay with a wide range of disulfiram concentrations (0.625 µM to 100 µM). Later, we confirmed the preliminary results by comparing the antimicrobial effect of disulfiram to that of doxycycline and used reliable quantitative methods to establish the bactericidal activity [33,34]. Disulfiram in both soluble forms (DMSO or cyclodextrin) inhibited the growth of *B. burgdorferi* strain B31 MI at an MIC^90^ range of 0.74 to 2.97 µg/mL in case of log-phase cultures (~94%) and at 1.48 µg/mL in case of stationary phase cultures (~90%), (Figure 1 and Figure 2), with MBC varying from 1.48 µg/mL to 2.97 µg/mL for log and stationary phase cultures. The immediate deceleration in log and stationary phases of *B. burgdorferi* growth at the low dose of disulfiram treatment is attributed to the rapid cleavage of disulfiram by thiophilic residues in intracellular cofactors (e.g., coenzyme A reductase) [22], enzymes (e.g., thioredoxin) [23], metal ions (e.g., zinc and manganese) [25], and cofactors of *B. burgdorferi*, which are hypothesized to instigate an abrupt halt in *B. burgdorferi* metabolism, thereby killing *B. burgdorferi*. A similar mechanism of action is proposed for pathogens like *Giardia*, *Bacillus*, drug-resistant *Mycobacterium*, and multidrug-resistant *Staphylococcus* [19,20,35]. Disulfiram drug efficacy has been enhanced to a maximum level with the concentration range from 2.5 µM to 10 µM (Figure 1 and Figure 2), above which the effect is diminished (25 µM to 100 µM), atypical bell-shaped curve. We have also found that more than 1000 citations of molecules demonstrated a similar efficacy pattern to ours in numerous previous studies [36]. Anticancer drugs such as fulvestrant, sorafenib, and crizotinib, have critical aggregation concentrations (CACs) of 0.5−20 μM. Below their respective CACs, these drugs exist in a classic monomeric form where, at sufficiently high (monomeric) concentrations, they are toxic to cells; in contrast, above their CACs, these drugs form colloidal aggregates that are substantially less cytotoxic in cell assays [37,38]. In addition, it has been reported that disulfiram producing a biphasic cytotoxic response in some breast cancer cell lines [39], human tumor cell lines [40], and neuronal cells [41]. In these cell lines 1 μM disulfiram was toxic involving consistent loss of cell viability. These effects disappeared at 10 μM where morphology was comparable to diluent controls; then, the increase of the disulfiram concentration to 100 μM restored the toxic 1 μM phenotype. These studies prompted us to perform DLS and AFM imaging to investigate the aggregation formation of disulfiram in DMSO and CD. Our DLS results clearly indicates that disulfiram in DMSO and CD has a critical aggregation concentrations (CACs) of 10 μM and this was supported by the AFM imaging showing above 25 μM aggregation of disulfiram occurs. Based on these results we concluded that at high concentrations (25 μM to 100 μM) of disulfiram aggregate into large particles that cannot diffuse across the cell membrane, and as their concentration rises they act as sinks for even the free monomer, leading to a bell-shaped concentration−response shown in Figure 1 and Figure 2. Further extensive study is underway to establish the possibility of target-based mechanisms.

Disulfiram is an oral medication that is approved by the U.S. Food and Drug Administration (FDA) for the administration of up to 500 mg daily [42]. Pharmacokinetic studies in humans have shown that disulfiram has a half-life (t_1/2_) of 7.3 h and a mean plasma concentration of 1.3 nM, although significant intersubjective variations are noted [43]. The toxicity of both disulfiram and its metabolites have also been broadly investigated in cell and animal studies, which yielded no evidence for teratogenic, mutagenic, or carcinogenic effects [44]. DMSO is toxic at a low dose in vivo [45] so, we have used non-toxic cyclodextrin [46,47] as a solubilizing agent for disulfiram in vivo studies. Based on these observations, we conducted our preliminary in vivo mouse efficacy studies by administering (I.P.) low dose of disulfiram (i.e., 10 mg/kg of body weight) to infected C3H/HeN mice for 5 days and found that these mice were not able to clear the *B. burgdorferi* from tissues (unpublished data). As shown in the current study, however, when we repeated in vivo C3H/HeN mouse efficacy studies by administering (I.P.) 75 mg/kg of body weight disulfiram to infected mice for 5 days, all infected mice have either reduced or cleared the bacteria in most of the tissues at 21 and 28 post infection (Table 1 and Table 2 and Figure 3). C3H mice develop bradycardia and tachycardia beginning on day 7 through 60 days after *B. burgdorferi* inoculation and triggers severe inflammatory responses particularly in C3H mice on days 15 to 21 post infection [28]. Thus, we have chosen the C3H/HeN mouse model for our efficacy studies and day 14 or day 21 post infection as time points for antibiotics treatments. Lyme carditis, a macrophage-mediated pathology, is not directly influenced by *B. burgdorferi* specific antibodies, but by inflammatory micro environment coming from pro-inflammatory Th1 cytokines (IL-1β, TNF, and IFN-γ), anti-inflammatory Th2 cytokine (IL-10), and other M1/M2 macrophage-polarizing factors such as iNOS and NOS2 derived from macrophages and T cells [48,49,50]. Similarly, chemokines (e.g., MIP-2, KC, and RANTES) preferentially attract monocytes and lymphocytes significantly contributing to the inflammation and tissue damage in Lyme disease [51]. We have shown that in disulfiram-treated mice groups there is a significant reduction in the infiltration of leucocytes in the heart wall and no inflammation (inactive carditis) compared to the doxycycline-treated group (active mild carditis) and PBS infected group (active severe carditis) at day 21 or day 28 post infection (Figure 3). This implies that disulfiram treatment reduced the inflammatory microenvironment by reducing the inflammatory chemokines (MIP-2 and RANTES), and cytokines (IL-10, IL-1β, TNF, and IFN-γ) and further reduces the disease severity in the heart. Macrophage phenotype is flexible, and once the infection is cleared and a more anti-inflammatory environment is created, macrophages switch to a pro-resolution M2 phenotype [52]. Henceforth, in disulfiram-treated mice, the level of NOS2 (M2 polarizing factor) was elevated compared to those of the doxycycline-treated and PBS infected groups at day 21 or day 28 post infection (Figure 4). However, the underlying mechanism involved in differential expression of chemokines and cytokines and their effect on disease severity needs to be further investigated.

In addition, our study found a lower bacterial burden in the ear, heart and bladder of disulfiram-treated mice when compared to PBS-treated infected mice at 21 days post infection. This result indicates that the disulfiram administration has promoted the antibody-mediated killing early in the infection, and thus it not only limited the *B. burgdorferi* colonization in tissues but also altered the development of adaptive immune response, which is aligned to the reduction of tissue inflammation as observed in heart samples [53,54]. In fact, *B. burgdorferi* infection leads to strong and sustained IgM response and delayed development of long-lived antibody and B cell memory [55]. So, disulfiram-treated mice might have accelerated long lived antibody and B cell memory development, which resulted in statistically lower amount of Bb specific IgM, total IgM, IgG and IgG1 at day 21 post infection (Figure 5 and Appendix A). A similar pattern with a statistically lower amount of both Bb specific IgM and IgG was observed for disulfiram treated mice on day 28. On the other hand, at day 28 post infection, disulfiram-treated mice have higher amounts of total IgM, IgG1 and IgG3 isotypes similar to the saline-treated mice, which all bind to C1q and activate the classical pathway, whereas IgG2a and IgG2b bind to the Fc receptor [56]. As such, it is likely that those immuno-complexes formed with C1q-binding antibodies cannot be opsonized by the complement system during infection due to the absence of C1q, thus fail to be engulfed by phagocytes and accumulated within the circulation system. We have not further studied Bb-IgG subsets in detail because the current study involves short term experiments (21 and 28 days post infection are last time points). Therefore, our data indicate that memory to *B. burgdorferi* infection may have not be formed until late during infection. Moreover, to what extent long-lived plasma cells contribute to immune protection remains to be studied, which is the subject of our future study using chronic Lyme arthritis model. Lymphoadenopathy observed during Lyme borrreliosis is caused by a massive increase in lymph node cellularity triggered by the accumulation of live *B. burgdorferi* spirochetes into the lymph nodes. This increase in cellularity is caused by accumulation of CD19+ B cells [30]. Disulfiram treatment alleviates lymphoadenopathy by reducing the percentage of CD19+ B cells in day 28 post infected mice (Figure 6). An important function of CD4+ T cells is their ability to enhance antibody-mediated immunity by driving affinity maturation and the development of long-lived plasma cells and memory B cells [57]. However, it appears that the response of protective B cells to *B. burgdorferi*, a highly complex pathogen expressing many immunogenic surface antigens, is confined to T-independent antibody responses alone. Even though disulfiram-treated mice induced an increase in percentage of CD3+ CD4+, Naïve, effector and memory T cells, further studies are needed to understand the role of these increased T cells in disease resolution and bacteria clearance.

In summary, the disulfiram drug not only successfully cleared the bacteria but also suppressed the inflammatory responses in heart tissues of C3H/HeN mice on day 28 post infection. Furthermore, disulfiram reduced antibody titers followed by nullifying lymphoadenopathy. The preclinical data offered here is beneficial in ascertaining the effectiveness of disulfiram and aids in performing future mechanistic and translational research studies. Moreover, the inhibitory effects of disulfiram on borrelial metabolism leave a room to exploit multiple mechanisms that can be used as therapeutic targets. Although the results from our in vivo study cannot be extrapolated directly to clinical practice at this point, we strongly believe they form a strong basis for future follow-up studies, and promote the development of effective formulations of disulfiram for clinical management of Lyme disease.

## 4. Materials and Methods

### 4.1. Culturing and Growth Conditions of B. burgdorferi B31 MI

*Borrelia burgdorferi sensu stricto* low passage strain B31 MI was (obtained from Luciana Richer, US biologics, Memphis, TN, USA) used for MIC tests and all infection studies in C3H/HeN mice. Bacteria cultures were started by thawing −80 °C glycerol stocks of *B. burgdorferi* (titer, ~10^7^ CFU/mL) and diluting 1:40 into fresh Barbour-Stoner-Kelly (BSK) complete medium with 6% rabbit serum followed by incubating at 33 °C. After incubation for 4–5 days log phase, and 8–9 days stationary-phase *B. burgdorferi* culture (~10^6^ borrelia/mL) were transferred into a 48-well plate for evaluation with the drugs.

### 4.2. Drug Formulations

The disulfiram (Sigma, St. Louis, MO, USA) stock solution (50 mM) was made by dissolving in sterile 30% hydroxypropyl β-cyclodextrin (Sigma) and also another disulfiram stock solution (20 mM) was made by dissolving in sterile 100% DMSO (Tocaris bioscience,Bristol, UK). A stock solution of 100 mM of doxycycline (as a positive control) was made by dissolving the doxycycline powder in ultra-pure Milli Q water. All drug stocks were passed through 0.22 μm filters (Millipore-Sigma, St. Louis, MO, USA), used within 72 h of preparation and were not subject to freezing temperatures. Working solutions were made by mixing the desired volume of stock solutions in the desired volume of ultra-pure MilliQ water. Furthermore, the vehicle for hydroxypropyl β-cyclodextrin (cyclodextrin) and DMSO controls were made similarly and it is important to note that the vehicle controls were identical to the test formulation in every single aspect except for the active ingredient. This measure was strictly followed for vehicle control wherever used in the entire study.

### 4.3. In-Vitro Testing of Antibiotics by Microdilution and Dark Field/Fluorescent Methods

A standard microdilution method was used to determine the minimum inhibitory concentration (MIC) of the antibiotics tested before [58]. Approximately, 1 × 10^6^
*B. burgdorferi* (log and stationary phase respectively) were inoculated into each well of a 48-well tissue culture microplate containing 900 μL of BSK medium per well. The cultures were then treated with 100 μL of each drug at varying concentrations ranging from 0.625, 1.25, 2.5, 5, 10 and 20 μM. Control cultures were treated with respective vehicles, and all experiments were run in triplicate. The well plate was covered with parafilm and placed in the 33 °C incubator with 5% CO_2_ for 4 days. Spirochetes proliferation was assessed using a bacterial counting chamber (Petroff-Hausser Counter) after the 4–5 days incubation followed by dark-field microscopy respectively.

#### SYBR^®^ Green I/PI by Fluorescent Microscopy

As a confirmation test, the SYBR Green/PI method was used for cell growth by directly counting live and dead bacteria by fluorescent microscopy. To evaluate live and dead cells, standard SYBR Green I/propidium iodide (SYBR Green I/PI) was performed as previously described [34,59]. To 1 mL of sterilized distilled water, 10 μL of SYBR Green I (10,000× stock, Invitrogen, Grand Island, NY, USA) and 30 μL of propidium iodide (Thermo Scientific, Waltham, MA, USA) were briefly mixed. The staining mixture (10 μL) was added to all the wells containing *B. burgdorferi* and was incubated in the dark for 15 min. The standard equation was determined from 1 × 10^6^ cells (logarithmic phase) and 5 × 10^6^ cells (stationary phase). A live and dead population was prepared. For the dead cell population, the cells were killed by adding 300 μL of 70% iso-propyl alcohol (Fisher Scientific, Santa Clara, CA, USA). We counted (200×) live (SYBR green) or dead (PI) bacteria cells in each condition of control or treatment (took at least 6 fields per condition) by fluorescent microscope. We combined the averages of live or dead bacteria cells per each condition to obtain total cells per condition. To generate a standard curve, different ratios of live and dead cell suspensions (live:dead ratios = 0:10, 2:8, 5:5, 8:2, 10:0) were added to the wells and stained as aforementioned in methods. Using the least square fitting analysis, the regression equation was calculated between the percentage of live bacteria and green/red fluorescence ratios. The regression equation was used to calculate the percentage of live or dead cells in each sample. Also, images of the treated sample were taken using fluorescent microscopy.

To further determine the minimum bactericidal concentration (MBC) of the antibiotics tested (the minimum concentration beyond which no motile spirochetes can be sub cultured after a 21 days incubation period), wells of a 48-well plate were filled with 1 mL of BSK medium and 20 μL of antibiotic-treated spirochetes were added into each of the wells. The well plate was wrapped with parafilm and placed in the 33 °C incubator with 5% CO_2_ for 21 days. After the incubation period, the plate was removed and observed for motile spirochetes in the culture by dark-field and further cell proliferation was assessed using the SYBR Green I/PI assay fluorescence microscopy. All these experiments were repeated at least three times. Statistical analyses were performed using Student’s *t*-test.

Semisolid plating method: We performed a semisolid plating procedure as described [60]. The 2× BSK-II medium was prepared in the following manner. To the 500 mL of CMRL-1066 medium: 50 g of bovine serum albu-min (Sigma), 5 g neopeptone (BD), 6.6 g HEPES acid (Sigma), 0.7 g sodium citrate (Sigma), 5 g glucose (Sigma), 2 g yeastolate (BD), 2.2 g sodium bicarbonate (Fisher), 0.8 g sodium pyruvate (Sigma), 0.4 g N-acetyl-glucosamine (Sigma) were added and mixed thoroughly. Finally, the pH of the medium was adjusted to 7.6 and filtered through 0.2 µm filter units. For plating the medium is mixed in the following way. The 250 mL of 2× BSK-II medium prewarmed at 55 °C was mixed with 250 mL 1.75 mL of agarose (55 °C) and 35 mL sterilized rabbit serum and equilibrated to 55 °C. Then 8 mL of equilibrated BSK-II medium was dispensed into 100-mm Petri dishes as bottom agar and allowed to solidify. Finally, the sample was resuspended in 0.5 mL fresh BSK-II medium and mixed with 8 mL of BSK-II agarose medium (55 °C) and poured as a top agar. The plates were incubated in the incubator with 5% CO_2_ at 35 °C for a minimum of 21 days. The white visible colonies were counted after 21 days for the analysis. Finally, semisolid plating was chosen to obtain the exact count of the growing borrelial colonies as colony forming units (CFU).

### 4.4. Dynamic Light Scattering

Since the disulfiram is insoluble in water, the stock solutions of 1M disulfiram were prepared either in DMSO or in 30% (*w/v*) hydroxypropyl β-cyclodextrin (CD). Disulfiram was then diluted in bovine serum albumin (BSA) solution to obtain disulfiram concentration 0.125 µM, 0.25, 0.5, 10, 25, 50, and 100 µM, and 5% (*w/v*) BSA in the final solution for DLS. The measurements were obtained from the Brookhaven 90-Plus particle size analyzer (Brookhaven instruments corporation) at an angle of 90° with 10% dust cutoff filter. The results represent an average of three measurements.

### 4.5. Atomic Force Microscopy

Atomic force microscopy (AFM) samples were prepared from drugs disulfiram-CD and disulfiram-DMSO solutions of respective concentrations (100 μM, 25 μM, 10 μM and 5 μM) on clean silicon wafers that were plasma-treated to increase hydrophilicity. Then, 10 µL droplets were deposited, spreading for most of the surface of 1 cm^2^ wafers and were quickly dried in a desiccator under vacuum to minimize additional aggregation due to local increase in concentrations. AFM imaging was performed with NX-10 AFM (Park Systems, Suwon, Korea) operating in non-contact mode with Micromasch NCS15 AL BS tips (NanoandMore, Watsonville, CA, USA) at 0.8 Hz with 256 pixels per line.

### 4.6. Animal Experiments Ethical Statement

All mice were maintained in the pathogen-free animal facility according to animal safety protocol guidelines at Stanford University under the protocol ID APLAC-30105. All experiments using animals were conducted according to the Administrative Panel on Laboratory Animal Care guidelines at Stanford University.

### 4.7. In Vivo Testing of Drugs in Immunocompetent C3H/HeN Mice

Four-week-old female C3H/HeN mice were purchased from Charles River Laboratories, Wilmington, Massachusetts. The mice (5 weeks) were infected subcutaneously close behind the neck with 0.1 mL BSK medium containing log phase 10^5^
*B. burgdorferi* B31 MI. For in vivo studies, we used only disulfiram soluble in cyclodextrin. On the 14 and 21 days post Bb infection, the mice were intraperitoneally administered a daily dose of drugs, disulfiram (75 mg/kg) and doxycycline (50 mg/kg) for 5 consecutive days (Figure 3A). After 48 h of the last dose of administering compounds, both groups (day 21 and day 28 post Bb infection) of mice were terminated and their urinary bladders, ears, and hearts were collected. The DNA was extracted from the urinary bladder, ear and heart. The absence of *B. burgdorferi* marked the effectiveness of the treatment in these organisms. Quantification of important pro/anti-inflammatory immune marker transcripts and histopathology of heart was also done. At termination on day 28 post infection, spleen and peripheral lymph nodes (axillary, brachial, cervical and inguinal) were also collected for immunophenotyping.

### 4.8. Quantitative (Q-PCR) and Real-Time PCR (RT-PCR) Analysis

Urinary bladder, ear punches, heart bases were homogenized, and DNA was extracted using the NucleoSpin tissue kit according to the manufacturer’s instructions (Macherey-Nagel, Düren, Germany). Q-PCR from the above tissues were performed in blinded samples using *B. burgdorferi* Fla-B gene-specific primers and a probe. These primers were listed as follows: Fla-B primers Flab1F 5′-GCAGCTAATGTTGCAAATCTTTTC-3′, Flab1R 5′-GCAGGTGCTGGCTGTTGA-3′ and TAMRA Probe 5′-AAACTGCTCAGGCTGCACCGGTTC-3′ according to the published protocol. Reactions were performed in duplicate for each sample. Results were plotted as the number of Fla B copies per microgram of tissue. The lower limit of detection was 10 to 100 copies of *B. burgdorferi* Fla-B DNA per mg of tissue. In addition to standard laboratory measures to prevent contamination, negative controls (containing PCR mix, Fla-B primers, probe, and Taq polymerase devoid of test DNA) were included.

Total RNA was extracted from tissues using the RNeasy mini kit (Qiagen, Germantown, MD, USA) and reverse-transcribed using a high-capacity cDNA reverse transcription kit (Invitrogen, USA). cDNA was subjected to real-time PCR using primer and TAMRA probes (Stanford Protein and Nucleic acid Facility) previously described [61]. PCR data are reported as the relative increase in mRNA transcript levels of CxCL1 (KC), CxCL2 (MIP-2), CCL5 (RANTES), IL-10, TNF-α, IFN-γ, and iNOS/NOS2 normalized to respective levels of GAPDH.

## 5. Histopathology

For histopathology, heart samples were fixed in 10% formalin buffer followed by staining of vertical histological sections with hematoxylin and eosin dye. Heart tissues were assessed for inflammation by microscopic examination at intermediate (10×), and high (40×)-power magnification and were scored for severity of inflammation (carditis, vasculitis) according to the percentage of inflammation at the heart base upon examination at low power (10×). Scores of 0 (none), 1 (minimal; less than 5%), 2 (mild; between 5% and 20%), 3 (moderate; between 20% and 35%), 4 (marked; between 35% and 50%), and 5 (severe; greater than 50%) were assigned for the severity of inflammation [28,62]. Myocarditis consisted of focal or diffuse interstitial infiltrates of mononuclear leukocytes in the myocardium. The microscopic photographs were captured on an Olympus CX-41 microscope (Olympus, Tokyo, Japan). The images are shown at 10× and 40× magnification

### 5.1. Quantification of Borrelia Specific and Total Immunoglobulins in Serum by ELISA

The wells of enzyme-linked immunosorbent assay (ELISA) plates (MaxiSorp; Thermo Fisher, Waltham, MA, USA) were coated with 100 μL of heat-killed *sonicated Borrelia* bacteria (5 mg/mL) in 100 mM sodium carbonate (pH 9.7). The protein concentration was determined using the Lowry assay, and 1 to 5 μg of protein was used to coat each well of the plate. Anti-*Borrelia* IgM and IgG levels were measured using serum from infected and uninfected mice and horseradish peroxidase-conjugated anti-mouse immunoglobulin secondary antibodies (1:5000; Invitrogen, USA). Readings were done at 450 nm using a SpectraMax microplate reader (Molecular Devices, Sunnyvale, CA, USA). Quantification of total mouse immunoglobulin concentration IgA, IgM, IgG, IgG1, IgG2a, IgG2b, and IgG3 in mouse serum was done using Ready-Set-Go ELISA kits (Invitrogen, USA).

### 5.2. Flow Cytometry

Single-cell suspensions of lymphoid tissues were prepared as described [61], live/dead cell viability Pacific blue stain was used to eliminate dead cells followed by single cells separation from doublets by FSC-A vs FSC-H plots (Figure 6a). Cells were incubated in Fc blocking for 15 min at 4 °C in staining buffer and incubated with the appropriate marker for surface staining in the dark for 30 min at 4 °C. The following are the surface lineage markers used: CD3 clone 17A2 conjugated with fluorescein isothiocyanate (FITC) (1:200; BD Biosciences), CD4 clone RM4-5 conjugated with phycoerythrin (PE) (1:150; BD Biosciences), CD8 clone 53–6.7 conjugated with allophycocyanin (APC)-Cy7 (1:150; BD Biosciences), CD62L clone MEL-14 conjugated with PE-Cy7 (1:150; BD Biosciences), CD44 clone IM7 conjugated with APC (1:150; BD Biosciences), and B cells was CD-19 conjugated with PercP Cy5.5 (Tonbo Biosciences, San Diego, CA, USA) were described previously [61]. Cells were acquired on a BD-LSR II flow cytometer and data analyzed using Flow Jo software. Fluorescence-minus-one (FMO) controls were used for gating analyses to distinguish positively stained from negatively stained cell populations. Compensation was performed using single-color controls prepared from BD Comp Beads (eBioscience) for cell surface staining. Compensation matrices were calculated and applied using FlowJo software (TreeStar). The biexponential transformation was adjusted manually when necessary. Cells were gated based on their forward and side scatter profiles. Data analyses were performed with the acquisition of a minimum of 100,000 events

## 6. Statistics

Data analysis was done using Graph Pad Prism software. Single comparisons within uninfected or drug-treated groups and infected groups were analyzed with a two-tailed paired *t*-test, with unpaired *t*-test with Welch’s correction and with multiple *t*-tests. α = 0.05 for all tests. * *p* < 0.05, ** *p* < 0.01, *** *p* < 0.001, **** *p* <0.0001.

## Figures and Tables

**Figure 1 antibiotics-09-00633-f001:**
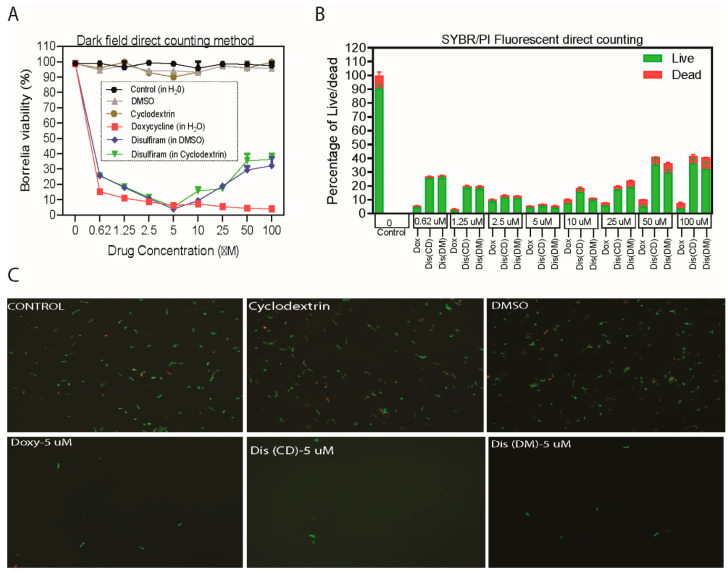
Evaluation of the borreliacidal activity of disulfiram (in DMSO and cyclodextrin) with doxycycline as control. A 4-day-old, *B. burgdorferi* log phase culture of *B. burgdorferi* was incubated for four days with disulfiram (Dis-DM), disulfiram (Dis-CD) and doxycycline (Doxy) at the same drug concentrations of 100 μM to 0.625 μM respectively. After a five-day incubation, bacteria cell viability was assessed by (**A**), by direct counting using dark field microscopy and (**B**), by SYBR Green-I/PI assay using fluorescent microscopy. (**C**), Representative images were taken using SYBR green-fluorescent stain (live organisms) and propidium iodide red-fluorescent stain (dead organisms) at 200× magnification. All these experiments were repeated at least three times (*n* = 6). Error bars represent standard errors.

**Figure 2 antibiotics-09-00633-f002:**
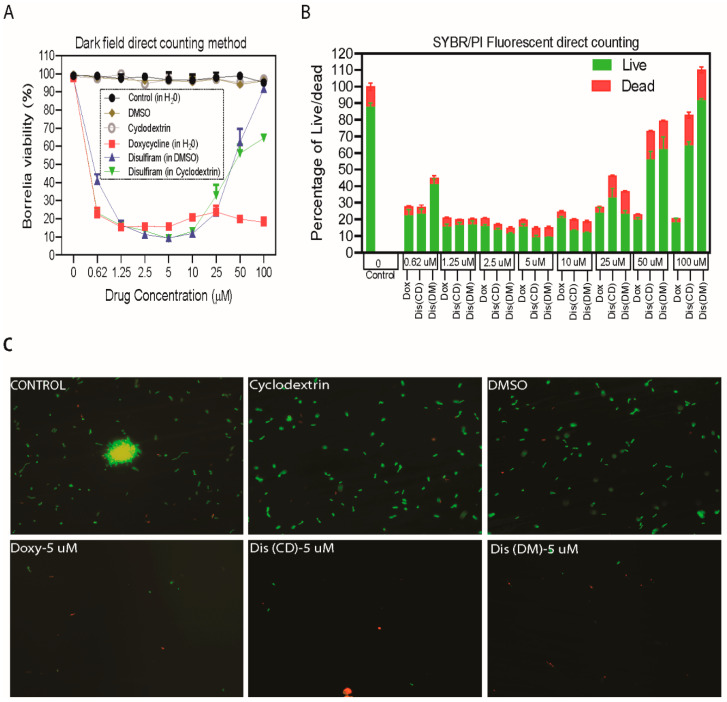
Evaluation of the activity of disulfiram (in DMSO and cyclodextrin) with doxycycline as control. An eight-day-old, *B. burgdorferi* stationary phase culture of *B. burgdorferi* was incubated for four days with disulfiram (Dis-DM) disulfiram, (Dis-CD) and doxycycline (Doxy) at the same drug concentrations of 100 μM to 0.625 μM respectively. After a five-day incubation, bacteria cell viability was assessed by (**A**), by direct counting using dark field microscopy and (**B**), by SYBR Green-I/PI assay using fluorescent microscopy. (**C**), Representative images were taken using SYBR green-fluorescent stain (live organisms) and propidium iodide red-fluorescent stain (dead organisms) at 200× magnification. All these experiments were repeated at least three times (*n* = 6). Error bars represent standard errors.

**Figure 3 antibiotics-09-00633-f003:**
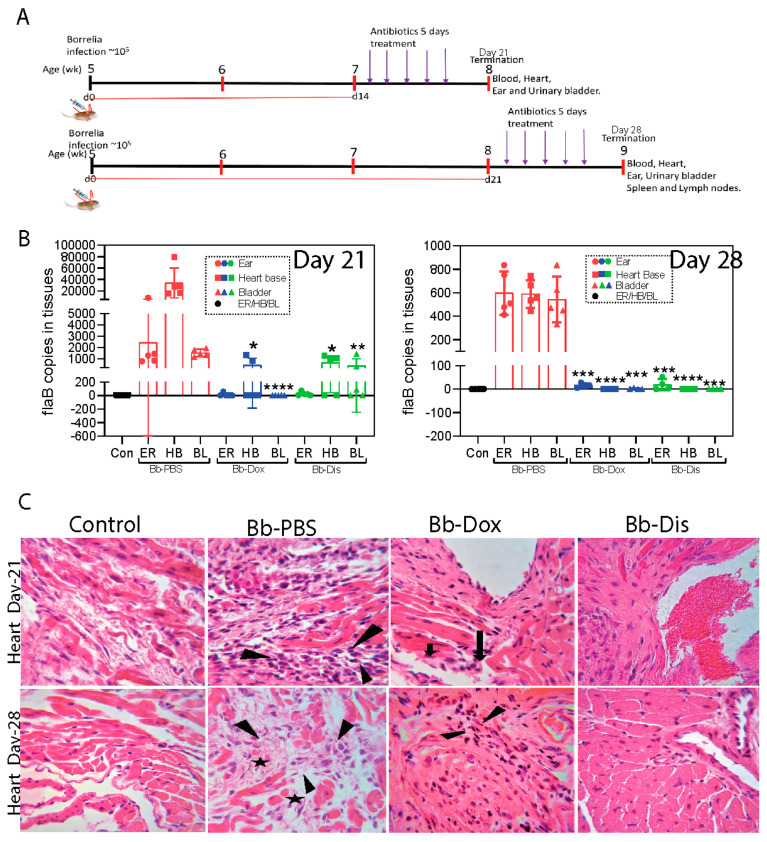
Borrelia loads in various tissues after C3H-HeN mice infection followed by disulfiram or doxycycline antibiotics treatment. (**A**) Antibiotic treatment/borrelia infection schedule: groups of 5 weeks old C3H mice (*n* = 30) were infected subcutaneously above the shoulders with mid log-phase 10^5^
*B. burgdorferi.* Infected groups were received intraperitoneal antibiotics doxycycline (*n* = 5), disulfiram (*n* = 5) and PBS (*n* = 5) at two different time points: day-14 post infection and day-21 post infection. Uninfected groups of mice were kept as controls (*n* = 10). (**B**) Necropsy at the end of day-21 and day-28, ears, hearts and urinary bladders were collected for determination of the number of *Borrelia* flab per ul of sample by qPCR. (**C**) Photomicrographs (40×) of hematoxylin and eosin-stained heart sections; arrows depict the mono nuclear leucocyte infiltrates. Statistics by unpaired *t*-test with Welch’s correction between drug-treated group versus infected group. * *p* < 0.05, ** *p* < 0.01, *** *p* < 0.001, **** *p* <0.0001.

**Figure 4 antibiotics-09-00633-f004:**
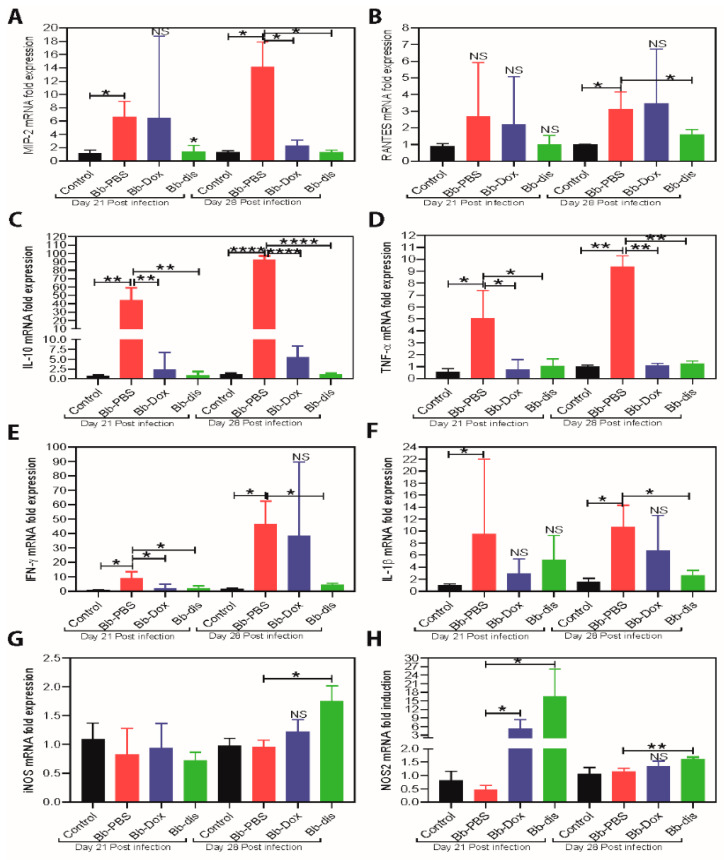
Measurement of immunomodulators in the hearts of C3H/HeN mice with or without antibiotics treatment by RT-PCR. (**A**–**H**) RT-PCR of pro-inflammatory transcripts (MIP-2, RANTES, TNF-α, IFN-γ and IL-1β) and important protective immunoregulatory transcripts (IL-10, iNOS and NOS-2) in the heart. Statistics by unpaired *t*-test with Welch’s correction between control versus infected and also between drug-treated group versus infected group. * *p* < 0.05, ** *p* < 0.01, **** *p* < 0.0001. NS means not significant.

**Figure 5 antibiotics-09-00633-f005:**
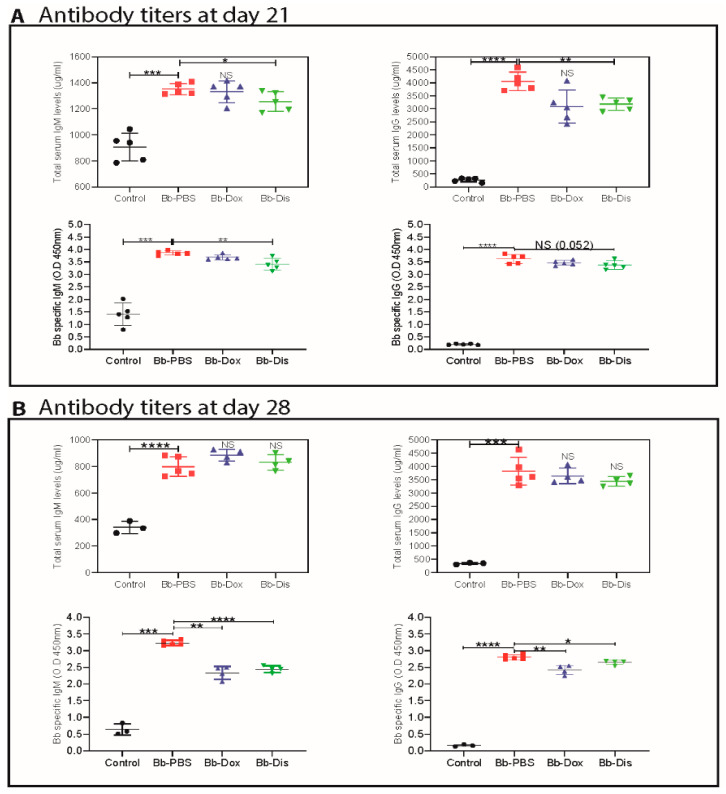
Antibody response in the serum with and without antibiotics treatment which were quantified by ELISA. Total and Bb specific concentration of IgM, IgG, antibodies (**A**) day 21 post- and (**B**) day 28 post-infection. Statistics: unpaired *t*-test with Welch’s correction between controls versus infected and between drug-treated group versus infected group. * *p* < 0.05, ** *p* < 0.01, *** *p* < 0.001, **** *p* < 0.0001. NS means not significant.

**Figure 6 antibiotics-09-00633-f006:**
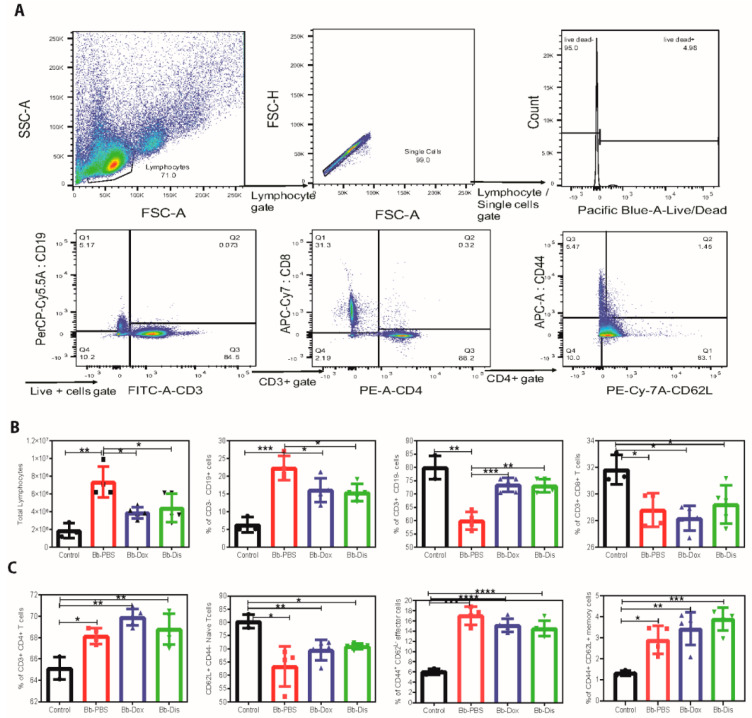
(**A**) Representative flow cytometry dot blot showing the gating strategy adopted for immuno-phenotyping of lymph nodes used in this study. (**B**,**C**) Percentage of B cells, T cells, CD8+ cytotoxic T cells, CD4+ helper T cells, naïve, effector and memory CD4+ T cells in lymph nodes. Flow cytometry analysis of immune cells isolated from peripheral lymph nodes from uninfected and infected mice treated with antibiotics. Cells were labelled with anti-CD19, anti-CD3, anti-CD4, anti-CD8, anti-CD44 and anti-CD62L lineage surface markers. Statistics unpaired *t*-test with Welch’s correction between control versus infected and between drug-treated group versus infected group. * *p* < 0.05, ** *p* < 0.01, *** *p* < 0.001, **** *p* < 0.0001. NS means not significant.

**Table 1 antibiotics-09-00633-t001:** In vivo efficacy of drugs against *B. burgdorferi* in C3H/HeN mice (on day 14).

No. of Mice Infected	Drug Name	Post 21 Days Ear Culture in BSK Medium	No. of *fla-b* DNA Copies/Ear	No. of *fla-b* DNA Copies/Bladder	No. of *fla-b* DNA Copies/Heart
1	Saline (No drug)	+	847	1761	15893
2		+	791	1422	28683
3		+	7851	1335	79512
4		+	1275	1136	16256
5		+	1447	1933	28125
1	Doxycycline	-	0	0	842
2		-	4	0	12
3		-	0	0	0
4		-	56	0	1325
5		-	10	0	0
1	Disulfiram	-	73	0	935
2		-	0	1439	0
3		-	0	0	1044
4		-	20	307	1297
5		-	20	89	0

After 14 days of *B. burgdorferi* infection, C3H/HeN mice were treated with following drugs once per day for 5 days (Doxycycline—50 mg/kg and Disulfiram—75 mg/kg). The whole DNA was extracted from urinary bladder, ear and heart and further analyzed data by qPCR. (+) Borrelia observed in 21 days post culture of ears in BSK medium. (−) No borrelia observed in 21 days post culture of ears in BSK medium.

**Table 2 antibiotics-09-00633-t002:** In vivo efficacy of drugs against *B. burgdorferi* in C3H/HeN mice (on day 21).

No. of Mice Infected	Drug Name	Post 21 Days Ear Culture in BSK Medium	No. of *fla-b* DNA Copies/Ear	No. of *fla-b* DNA Copies/Bladder	No. of *fla-b* DNA Copies/Heart
1	Saline (No drug)	+	477	625	447
2		+	509	469	663
3		+	412	837	534
4		+	835	448	745
5		+	753	331	556
1	Doxycycline	-	28	0	0
2		-	5	0	0
3		-	14	6	0
4		-	15	0	0
1	Disulfiram	-	0	0	0
2		-	21	0	0
3		-	0	0	0
4		-	51	0	0

After 21 days of *B. burgdorferi* infection, C3H/HeN mice were treated with following drugs once per day for 5 days (Doxycycline—50 mg/kg and Disulfiram—75 mg/kg). The whole DNA was extracted from urinary bladder, ear and heart and further analyzed data by qPCR.

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
