# Peer review of "Repurposing Disulfiram (Tetraethylthiuram Disulfide) as a Potential Drug Candidate against Borrelia burgdorferi In Vitro and In Vivo"

_antibiotics, 2020, doi:10.3390/antibiotics9090633_

Round 1

Reviewer 1 Report

The authors propose that disulfiram, an acetaldehyde dehydrogenase inhibitor, can be repurposed for the treatment of borreliosis. The need for this, according to the authors, is that a substantial proportion (10-20%) remain infected following treatment with first line defenses. This is a controversial area in the field, but the prevailing opinion among researchers, at least in the North American and Europe, is that antibiotic therapy is highly effective at killing the bacterium, but the symptoms are longer last due to dysregulated immunological responses in a subset of patients. Moreover, recently, Trautmann et al. (Front. Med [Lausanne] 2020) reported a high incidence of toxicities with treatment of late stage Lyme disease. For these reasons, this reviewer does not believe that the risks associated with use of disulfiram out weight any potential benefit. Thus, the clinical relevance of these studies is unclear.

The relevance of experiments testing aggregation of disulfiram ex vivo is unclear. According to values presented in the discussion, the drug concentration does not approach those used in vitro. Moreover, the dose administered to mice far exceeds the maximum recommended dose in humans.

PCR positivity is not an accurate readout for infection. Culturing of tissues in BSK medium should be performed to confirm the presence of live bacteria.

Live/Dead staining also is not an accurate readout for viability as it relies on perturbations in the outer membrane (for influx of PI). This may not occur even when the bacterium is no longer viable. The authors should include viable counts using semi-solid plating to more accurately assess viability.

Borreliacidal does not need to be italicized.

Author Response

Dear Reviewer,

please find the attached point to point response.

Thanks

Reviewer 2 Report

The authors are following up on a previous report that disulfiram can be used to control Borrelia burgdorferi growth.  This is an interesting observation as this drug is already FDA approved and could be repurposed for the treatment of Human Lyme Disease.  Indeed some are using this to treat various forms of Human Lyme disease. This report addresses this using in vitro and in vivo approaches and compares its effectiveness verses doxycycline, the current treatment standard.   Unfortunately, the current manuscript has a number of issues.

  1. The authors contrast the effects of disulfiram and doxycycline in vitro and this is presented in figures 1 and 2. The authors state in lines 113-115 that “However, doxycycline treatment had no significant effect on the cells in the stationary phase cultures…..” and again, lines 128-129 it is stated that “While doxycycline significantly reduced the viability of log phase cultures by 97%, it did not reduce the stationary phase viability of burgdorferi (Figure 2B).” It is unclear the basis for these statements as the data, examining the effects on stationary cultures, show to this reviewer that the effects of the two are similar if not identical – this needs to be explained. Perhaps there is a statistical difference between the two results, but no p-values are given. 

        Line 134 refers to Fig 3B – is that correct?

  1. Disulfiram fails to exert its antibacterial effects at higher does and this is the case whether DMSO or cyclodextrin is used as solvent.  The explanation for this is that the drug forms aggregates at these concentrations in either solvent and supplemental data is presented to support this.   What is not mentioned or discussed is that the aggregation properties of disulfiram seem to be quite different in the two solvents when measured by DLS (sFig 2) or AFM (sFig 5).  Might this mean that the loss of antibacterial effects might involve different mechanisms depending on the solvent used?

        It is not clear what the SDS studies shown in sFigs 3 & 4         add.  This should be either discussed more fully or removed. 

Note that line 145 refers to sFig1 – likely should be sFig 2.   Also, sFig 3B and 3C seems to be duplicates and sFig 3E has no data in it!!

  1. The authors indicate that disulfiram can lowers the bacterial burden and indicate that is more efficient that doxycycline. While it is clear that the former statement is true, its effectiveness over doxycycline seems overstated as the levels of detected Bb in the doxy group is quite low and are likely statistically insignificant. In fact, figure 3B seems to show that doxycycline is more effective early (Day 21).  
  2. The authors also indicate that disulfiram is more effective than doxycycline in limiting pathology in the heart but when the levels of carditis is measured (sFig 6 A & B), there seems to be no difference between the two treatments. Again, it seems the difference between the two treatments is exaggerated.
  3. It seems misleading to state that disulfiram reduced antibody titers in Bb infected mice as the authors are measuring total Ig in serum not an anti-Bb titer. It is surprising that one sees such dramatic changes in the levels of total Ig in infected mice and that both treatments impact this. Studies from the Baumgarth group has shown that Bb infection of mice stimulates a novel anti Bb response but have global changes in total Ig like this been reported in any Bb infected host? It is unclear how this may be happening and this needs to be discussed.   The treatments seem to lower these global levels, are they perhaps immunomodulatory? The measurement of the anti-Bb antibody response would be an important data set to include.   
  4. There is no discussion as to the meaning of the observed changes in immune cell subsets. Also, it is unclear how the flow cytometry analysis was conducted.  Was this done using polychromatic flow analysis or individual stains with PercP labeled antibodies?  This is unclear in the methods section.  In either case it is common to show representative flow histograms as supplemental information. 
  5. The authors frequently fail to place their results in the context of previous work in mouse models. For example, in line 360-363 the authors indicate that the B cell responses are confined to T-independent antibodies but do not provide a reference. In fact, TI antibodies are an important feature of the antibody responses to Bb but T cell dependent responses are as well. There are several studies from the Baumgarth lab and others using C3H mice that should be consulted.

Author Response

Dear Reviewer,

Please find the attached point to point responses.

Thanks

Round 2

Reviewer 1 Report

The authors have adequately addressed technical concerns. There are still quite a few text glitches that will need to be corrected.

Author Response

Dear Reviewer,

Please find the attached response document for your perusal.

Thanks

Hari Potula

Reviewer 2 Report

The authors have made considerable changes to the manuscript in response to the previous review.  

recommend rephrasing line 190-191: "These results showed disulfiram is similar to doxycycline in C3H/HeN mouse to restrict the......."

Author Response

Dear Reviewer,

Please find the attached response document for your perusal.

Thanks

Hari Potula

This manuscript is a resubmission of an earlier submission. The following is a list of the peer review reports and author responses from that submission.

Round 1

Reviewer 1 Report

Unfortunately I was unable to review any of the mouse work as none of the results correlated to the figures/ tables. Therefore, my comments will be directly purely at the cell culture, modelling and DLS experiments.

Cell culture-

There is no discussion regarding the difference between the 4 day and 8 day results. This is a huge hole as it looks like the longer these cells are treated, the less effective the drug is? Also looks to be significantly more dead cells at the higher concentrations of Dis, is this due to higher levels of aggregation? Or is Dis toxic?

Figure 1 has 3 panels, only two described in legend. Fig 1B – difficult to see the percentages and if there is any difference. Could this be zoomed in on so easier to see. Also not clear what the control is and why this is above 100%. Please explain.

DLS -

Supp Fig 1 – needs labeling. What is the red? What is the green? What are the dots? Since cells were grown in BSK medium with 6% rabbit serum – why were the same buffers not used for DLS? You can’t really conclude the same results using different buffers?

Modelling -

Please describe how similar your templates were to your sequence.

Did you only use the Ramachandran plot to assess model quality? If so, could you please incorporate more thorough assessment for example, look at the swissmodel assessment tools (all are freely available).

Fig 3 – very difficult to see labels etc. I would suggest turning off hydrogens (at least non-polar ones) and/or creating the labels in another program.

Minor typos:

Line 194 – Fig 3A is a modelling figure, do you mean 4A?

Line 76: “is is an electrophile”

Line 184 “aminoacids”

Author Response

We thank you for your valuable comments; we have made changes in the manuscript according to your comments and suggestions in manuscript, as described below with specific line numbers.

Reviewer 2 Report

The manuscript by Potula et al. presents results describing the activity of the FDA approved drug disulfiram against Borrelia burgdorferi. The research is topical, mostly well designed and carried out, and the results show that disulfiram is active against this bacterium in vitro and in a mouse model. The following comments should be addressed: 

(i) Lines 165-170: Are topoisomerase II and MgtE known targets of disulfiram? A clear rationale as to why these two proteins were selected over other proteins (e.g., involved in DNA synthesis) needs to be provided. Why only two proteins? Are the binding sites similar? A much deeper analysis of the drug-protein interface should be provided based on the modelling. 

(ii) Figure 3 is of rather poor quality and thus it is difficult to assess the non-covalent interactions between the drug and amino acids of the proteins. Additionally, Figures 3A and C provide limited information and could be removed, or at the very least highlight (by color for example) the amino acids interacting with the drug in Fig. 3A, and in Fig. 3C show the drug and amino acids interacting with the drug clearly. In panels B and D the different coloring of H-bonds and hydrophilic aa is not easily seen, and what colors are used as these are not given in the legend. In the description (lines 174 and 184-185) the amino acids involved in interacting with the drug are given. What role do these amino acids play in the protein, e.g., catalytic, functional, structural? A description of what role these residues play in the activity of the proteins should be provided in the context of the drug potentially disrupting the activity of these proteins. 

(iii) Tables 1 and 2 should be moved to the supplementary information. Additionally, there are a number of different types of interactions described in Tables 1 and 2 that are not mentioned in the text (lines 165-187). A more in-depth description (besides the mention of H-bonds) of how the ligand interacts with the proteins should be provided given the data presented in these tables. 

(iv) Were other ligand binding sites identified during the ligand-protein interaction modelling? If so, why are they excluded in this presentation, and if not, was this due to the method used and therefore limits the significance of the protein-drug modelling?

(v) Figure 1C and 2C could be reduced to one set of results for one drug because they show similar information, and the remaining 8 panels in each figure moved to the supplementary information.

(vi) Figure 1B and 2B show from the live/dead assay the percentage of live and dead cells in the population. Some explanation to how these % values were determined should be given in the methods section. Also, how many cells were counted for each sample and how were the standard errors determined (as stated in the legends). 

(vii) The C is missing in figures 1 and 2 of the legends, lines 724 and 734. 

(viii) The AFM results are not convincing (Fig. S2), partly because of issues mentioned by the authors. Scanning electron microscopy and/or NMR would likely give better direct/indirect readouts of colloidal/aggregate formation. The authors should consider carrying out such experiments (or others) to support the DLS results (Fig. S1), and thus the results presented in Figs. 1 and 2. Can the aggregates be disrupt by detergent? Examining the affect of a standard detergent(s) would support the conclusions drawn from these DLS experiments and should be performed. 

(ix) Abstract, line 21: what is a "significant percent"? If available, report what that value is because "most Lyme disease patients can be cured". This sentence is somewhat confusing as to what a significant percentage represents here. 

Author Response

(The authors gave the same response as above.)

Reviewer 3 Report

This research is a detailed study of exploring disulfiram as drug candidate against Borrelia burgdorferi, and authors have explored different aspect of this research. I have few suggestions and queries as below

  1. For burden in tissue assessment (Page 6, line 212) why only bladder, ear and heart were taken? Is there any specific consideration behind the study? As bladder and ear doesn’t come in vital organs?
  2. In some cases of Lyme disease hearing and balance loss was observed in previous literature, in present study do these infected mice shows hearing or balance loss? If yes, then do author have tested these parameters? And can this treatment reverse or stop it?
  3. Ear is a close compartment where delivery of drugs seems to be very tough. In this study it has been shown drug given intraperitoneally can reduce infection in ear, I was just wondering what could be the possible mechanism?
  4. There are some minor errors with spacing all over manuscript, Authors can re-read and correct accordingly.

Round 2

Reviewer 1 Report

Unfortunately the authors did not address my major concern "I was unable to review any of the mouse work as none of the results correlated to the figures/ tables." If I cannot review this work, I cannot suggest this article be published. 

Reviewer 2 Report

The authors have made sufficient attempts to address the comments raised. Please double-check that those references used in the response letter are also included in the manuscript to support the claims made. The English throughout the manuscript needs to be edited (professionally) to improve readability and clarity. 
